# Medroxyprogesterone Reverses Tolerable Dose Metformin-Induced Inhibition of Invasion via Matrix Metallopeptidase-9 and Transforming Growth Factor-β1 in KLE Endometrial Cancer Cells

**DOI:** 10.3390/jcm9113585

**Published:** 2020-11-06

**Authors:** Dong Hoon Suh, Sunray Lee, Hyun-Sook Park, Noh Hyun Park

**Affiliations:** 1Department of Obstetrics and Gynecology, Seoul National University Bundang Hospital, 82, Gumi-ro 173 Beon-gil, Bundang-gu, Seongnam-si, Gyeonggi-do 13620, Korea; sdhwcj@naver.com; 2Cell Engineering for Origin Research Center, Ujeongguk-ro, Jongno-gu, Seoul 03150, Korea; sunray@naver.com (S.L.); hspark@cefobio.com (H.-S.P.); 3Department of Obstetrics and Gynecology, Seoul National University College of Medicine, 101 Daehak-ro Jongno-gu, Seoul 03080, Korea

**Keywords:** metformin, medroxyprogesterone, endometrial cancer, invasion, matrix metallopeptidase-9, transforming growth factor-β1

## Abstract

This study was performed to evaluate the anticancer effects of tolerable doses of metformin with or without medroxyprogesterone (MPA) in endometrial cancer cells. Cell viability, cell invasion, and levels of matrix metallopeptidase (MMP) and transforming growth factor (TGF)-β1 were analyzed using three human endometrial adenocarcinoma cell lines (Ishikawa, KLE, and uterine serous papillary cancer (USPC)) after treatment with different dose combinations of MPA and metformin. Combining metformin (0, 100, 1000 µM) and 10 µM MPA induced significantly decreased cell viability in a time- and dose-dependent manner in Ishikawa cells, but not in KLE and USPC cells. In KLE cells, metformin treatment alone significantly inhibited cell invasion in a dose-dependent manner. The inhibitory effect of metformin was reversed when 10 µM MPA was combined, which was significantly inhibited again after treatment of MMP-2/9 inhibitor and/or TGF-β inhibitor. Changes of MMP-9 and TGF-β1 according to combinations of MPA and metformin were similar to those of invasion in KLE cells. In conclusion, the anticancer effects of tolerable doses of metformin varied according to cell type and combinations with MPA. Anti-invasive effect of metformin in KLE cells was completely reversed by the addition of MPA; this might be associated with MMP-9 and TGF-β1.

## 1. Introduction

Uterine corpus cancer was found to be the 4th most common cancer in women in 2018; the estimated number of new cases was 63,230, accounting for 7% of all new cancer diagnoses in women [1]. Endometrial cancer constitutes the majority of uterine cancers. Despite multimodal treatment approaches, type I poorly differentiated endometrioid adenocarcinoma and type II cancers, including uterine serous papillary cancer (USPC) without estrogen receptor (ER) and progesterone receptor (PR) expression, have very poor prognosis unlike type I well-differentiated endometrioid adenocarcinoma, which expresses ER and PR. Among the systemic hormonal therapies considered for recurrent, metastatic, or high-risk disease, progestin is the most commonly used, mainly in the form of medroxyprogesterone acetate (MPA). However, clinical guidelines recommend that MPA may only be used for lower-grade endometrioid histology. This is based on previous reports that the highest response rates were noted in low-grade, ER-positive tumors of up to 55% [2,3]. In addition, long-term continuous use of progestin was known to cause a loss of effect of PR activation [2]. Therefore, development of a new treatment strategy for groups of cancer with poorer prognosis is urgent.

Recently, metformin, an oral biguanide anti-diabetic drug for type 2 diabetes, was shown to have significant anticancer activity and considered as a novel treatment option through drug repositioning [4], for several types of cancer including endometrial cancer [5,6,7]. However, it should be noted that almost all previous studies were conducted with supra-pharmacological concentrations (doses) of metformin, which were 10–100 times higher than maximally achievable therapeutic concentrations found in patients with type 2 diabetes mellitus [8]. Such levels exceed the maximum dose that could cause lactic acidosis, one of the most serious side effects of metformin. Any anticancer effect of metformin should be studied only in the condition of achievable therapeutic concentrations [8,9].

Another approach for the development of novel anticancer drug regimens is the use of drug combinations. Although hormonal therapy is currently recommended only for lower-grade endometrioid histology in clinical guidelines, there is evidence suggesting that several anticancer mechanisms of progesteronal agents could show significant effects in poorly-differentiated endometrioid adenocarcinoma, as well as USPC [2,10,11], particularly when combined with metformin [12]. The purpose of this study was to evaluate the anticancer effect of tolerable doses of metformin with or without MPA as well as its underlying molecular mechanisms in endometrial cancer cells.

## 2. Experimental Section

### 2.1. Cell Cultures

Three human endometrial adenocarcinoma cell lines were used: Ishikawa (type I well-differentiated, ER+/PR+), KLE (type I poorly differentiated, ER-/PR-), and USPC (type II serous papillary carcinoma, ER-/PR-) [13]. Ishikawa cells were purchased from the Japanese Collection of Research Bioresources cell bank (Osaka, Japan) and maintained in Dulbecco’s Modified Eagle Medium (DMEM) (Life Technologies, Carlsbad, CA, USA) containing 10% fetal bovine serum (FBS) (Hyclone, Logan, UT, USA), 50 µg/mL streptomycin, and 50 U/mL penicillin. KLE cells were obtained from the American Type Culture Collection (ATCC, Manassas, VA, USA). KLE was cultured in DMEM/F12 medium (Life Technologies) with 10% FBS and 0.5% P/S. USPC cells (USPC-ARK-1) were purchased from Dr. Alessandro Santin (Yale University, New Haven, CT, USA) and were cultured in Roswell Park Memorial Institute (RPMI)-1640 medium (Life Technologies) with 10% FBS, 2 mM/L glutamine, and 0.5% P/S. All the cells were cultured in an incubator at 37℃ under a humidified atmosphere containing 5% CO_2_.

### 2.2. Dose Setup of Metformin and MPA Treatments

The tolerable doses of metformin which could achieve a plasma concentration of around 1 mg/L was 500 mg twice/day [14]. Although the maximal approved total daily dose of metformin for treatment of diabetes mellitus is 2.5 g (35 mg/kg body weight) [8], slow but progressive increase of fasting lactic acid levels were noted during metformin treatment with multiple doses from 100 to 850 mg twice a day, suggesting that; the higher dose of metformin, the higher risk of lactic acidosis [14]. The therapeutic plasma concentrations of metformin measured in previous studies of type 2 diabetes ranged from 0.129 to 90 mg/L [9]. Therefore, 1 mM (129.2 mg/L) was set as a maximal concentration of metformin for our experiment, enabling the maximum achievable plasma concentration in a clinical setting without the risk of lactic acidosis. 

MPA dose of 10µM was set based on a study of Zhang et al. [12] which also had evaluated the anticancer effect of MPA and metformin combination in endometrial cancer cells. MPA 10µM was high enough to inhibit proliferation of Ishikawa cells at 48 h. However, it was too low to suppress progestin-resistant Ishikawa cells, which were considered to have similar characteristics with KLE cells in our study. Progestin resistance of progestin-resistant Ishikawa cells was overcome by the addition of metformin to MPA 10µM [12]. KLE and USPC cells were not expected to be susceptible to higher dose of MPA because of negative expression of ER and PR. Therefore, 10µM was set as an optimal dose of MPA which could show possible anticancer effects when combined with metformin in these cell lines. There was another study of progestin and metformin in endometrial cancer showed that 10µM MPA was the minimal dose that could significantly inhibit growth of RL95-2 cells (ER+/PR+) [15]. 

### 2.3. Cell Counting and Cell Survival Analysis

To measure cell growth rate, all cells were seeded in 12-well plates (Corning Life Sciences, New York, NY, USA) at 10,000 cells/cm^2^, and the cell number was counted at 24-h intervals until 96 h. For cell counting, the medium was removed from the cell culture plates, washed twice with phosphate buffer saline (PBS), and then treated with 0.25% trypsin for 5 min at 37 °C. The trypsin-treated cells were collected in a 15 mL tube, washed twice with the culture medium, and counted three times using the Adam-MC automatic cell counter (NanoEntek, city, Korea). Viable cells were more accurately measured using an advanced image analysis program of Adam-MC cell counter.

Cell survival analysis was performed to investigate the effects of metformin (Sigma-Aldrich, St. Louis, MO, USA) and/or MPA (Sigma-Aldrich) on endometrial cancer cell lines. Cells were seeded into a 96 well plate (Ishikawa 5 × 10^4^ /cm^2^, KLE 2 × 10^4^/cm^2^ and USPC 3 × 10^4^/cm^2^). The next day, cells were treated with 100 µM and 1 mM of metformin, with or without 10 µM of MPA after media change. Then, survival rates of cells were analyzed after 24 h and 48 h of drug treatment using Ez-Cytox (DoGen Co., city, Korea), a water-soluble tetrazolium salt assay kit. The assay was performed according to the supplier protocol (http://www.dogenbio.com/shop/item.php?it_id=1490923054). The results were detected at 450 nm of absorbance.

### 2.4. Western Blot

The proteins collected from each cell sample were quantitated, subjected to 12% sodium dodecyl sulphate-polyacrylamide gel electrophoresis (SDS-PAGE), and then transferred to a nitrocellulose membrane. The membrane was subjected to blocking in PBS, containing 0.1% Tween20 (Sigma, St. Louis, MO, USA) and 5% skim milk (Invitrogen, Carlsbad, CA, USA), probed with primary antibodies, Glyceraldehyde 3-phosphate dehydrogenase (GAPDH) (14C10, Rabbit mAb), progesterone receptor-B (C1A2, Rabbit mAb), 5’ AMP-activated protein kinase *α* (AMPKα) (23A3, Rabbit mAb), phospho-AMPKα (Thr172, 40H9, Rabbit mAb) (Cell Signaling, Beverly, MA, USA), and erb-b2 receptor tyrosine kinase 2 (ErbB2) (Abcam, Cambridge, UK), and then reacted with peroxidase conjugated secondary antibody (Jackson Immuno Research, West Grove, PA, USA). Finally, target bands were visualized using SuperSignal chemiluminescent (ThermoFisher Scientific, Waltham, MA, USA). The immune-positive band was detected by ImageJ software [16], which was used to analyze the gray value of the protein expression. All protein quantifications were adjusted for GAPDH levels.

### 2.5. Cell Invasion Assay and ELISA

To perform invasion assays, we first coated matrigel (BD Science, San Jose, CA, USA) on a transwell membrane with 8 μm pores (Corning Life Sciences) at 37℃ for 2 h, and seeded 8 × 10^4^ cells/cm^2^ into the transwell membrane. The next morning, the cells were starved for 2 h in culture medium without fetal bovine serum (FBS). The outside of the transwell was replaced with medium containing 5% charcoal strip FBS (ThermoFisher Scientific, Waltham, MA, USA) to induce invasion with or without anti-cancer drugs (metformin and MPA) and each 10 μM inhibitor, MMP-2/9 inhibitor [17] (Sigma-Aldrich; Merck Millipore, Burlington, MA, USA) and TGF-β inhibitor (Tocris Bioscience, Bristol, UK), for 24 h at 37℃. The concentration of 10 μM of the two inhibitors was chosen from previous studies [17,18,19]. The next day, all the cells in the transwells were removed using cotton buds, and the transwells were inverted to stain the transferred cells with 0.2% crystal violet. The stained cells were de-stained with 2% SDS and the supernatant was transferred into new 96 well plate. The absorbance was measured at 560 nm.

For quantitative analysis of cell migration related proteins, the secretion levels of MMP-2 and -9 (R&D Systems, MN, USA) and TGF- β (R&D Systems) were checked using ELISA kits. First, all cells were plated at 9 × 10^4^ cells/cm^2^ into a 24 well plate (Corning Life Sciences, NY, USA) and starved for 2 h in culture medium without FBS. The anticancer drugs were then treated at various concentrations while the culture medium was exchanged with the complete medium. After 24 h, the cultures were collected without cells and analyzed. An ELISA was performed as per the supplier’s instructions (https://www.rndsystems.com/).

### 2.6. Statistical Analysis 

The statistical analyses were performed using GraphPad PRISM (GraphPad Software Inc., San Diego, CA, USA) and SPSS software (version 21.0; SPSS Inc., Chicago, IL, USA). Shapiro-Wilk test was used to check the distribution of data from three independent experiments and the test results confirmed that all data were normally distributed (*p* value > 0.05). Therefore, means of the two groups were compared using a two-tailed unpaired Student’s *t*-test. Bonferroni correction was performed for multiple testing correction and Bonferroni corrected *P*-values were used for statistical significance. Linear regression analysis was performed for estimating a trend of change. *P*-value < 0.05 indicated statistical significance.

## 3. Results

### 3.1. Cell Growth and Growth Inhibition by Tolerable Doses of Metformin and MPA in Endometrial Cancer Cell Lines

We found that USPC cells had the fastest growth rate among the three endometrial cancer cells during 96-h incubation, followed by Ishikawa and KLE cells (Figure 1A,B).

The MTT assay showed that treatment with metformin alone at ≤1000 µM for 48 h exerted significant inhibitory effects on the cell viability of Ishikawa, KLE, and USPC cells in a dose-dependent, but not in a time-dependent manner (Figure 1C–E). In Ishikawa cells, a combination of metformin (0, 100, 1000 µM) and 10 µM MPA induced a significant decrease in cell viability in a time- and a dose-dependent manner (linear regression: *p* < 0.05) (Figure 1C). Addition of 10 µM MPA to metformin significantly inhibited cell viability compared to metformin alone at each dose (0, 100, 1000 µM), respectively, in Ishikawa, but not in KLE and USPC cells.

### 3.2. Changes in Expression Levels of PR and p-AMPKα by a Tolerable Dose of Metformin and MPA in Endometrial Cancer Cell Lines

A significant level of endogenous expression of progesterone receptor-B (PR-B) was found in Ishikawa cells but not in KLE and USPC cells (Figure 2). In Ishikawa cells, metformin treatment alone induced the expression of PR-B in a dose-dependent manner, whereas metformin combined with 10 μM MPA inhibited PR-B expression in a dose-dependent manner. Expression of activated form of AMPKα, phospho-AMPKα (p-AMPKα), was inhibited by metformin treatment alone in a dose-dependent manner (0, 100, 1000 µM) in Ishikawa cells. However, p-AMPKα lost its dose-dependent pattern when Ishikawa cells were treated with a combination of metformin (0, 100, 1000 µM) and 10 µM MPA. In KLE and USPC cells, there were no significant changes in expression patterns of ErbB2, and AMPKα when treated with any of the doses of metformin and MPA. In AMPK/mammalian target of rapamycin (mTOR) pathway, high expression of p-AMPKα results in growth inhibition via inhibiting mTOR. Even though the expression of p-AMPKα was stronger when both metformin and MPA were used than when metformin was used alone in USPC cells (Figure 2), cell growths of the combination group were not significantly lower than those of metformin alone group (Figure 1E). In addition, there was no dose-dependent increase of p-AMPKα expression according to metformin doses (0, 100, 1000 µM) in USPC cells (Figure 1E).

### 3.3. Inhibition and Disinhibition of Cell Invasion by a Tolerable Dose of Metformin With or Without MPA in Endometrial Cancer Cell Lines

We further performed an invasion assay (Figure 3A), which showed that metformin treatment alone did not induce any significant changes in cell invasion in Ishikawa and USPC cells (Figure 3B,D). In KLE cells (Figure 3C), however, metformin treatment alone significantly inhibited cell invasion in a dose-dependent manner (1.31 ± 0.05, 0.94 ± 0.04, 0.83 ± 0.05 at 0, 100 µM, 1 mM, respectively; *p* < 0.0005). Treatment with MPA 10 µM alone significantly decreased the invasion of KLE cells compared to that of control cells (1.31 ± 0.05 vs. 1.10 ± 0.05; *p* < 0.005). Interestingly, the inhibitory effect of metformin alone on cell invasion was reversed when metformin was combined with 10 µM MPA (1.10 ± 0.05, 1.42 ± 0.18, 1.41 ± 0.26 at 0, 100, 1000 µM, respectively; *p* < 0.005) (Figure 3C). In Ishikawa cells, by contrast, a combination of 10 µM MPA with metformin exerted a significant inhibitory effect on cell invasion (0.93 ± 0.05, 0.76 ± 0.01, 0.69 ± 0.01, at 0, 10 µM MPA alone, 100 µM metformin and 10 µM MPA, respectively; *p* < 0.0005), although the inhibitory effect on cell invasion of MPA and metformin combination disappeared at a metformin dose of 1000 µM (0.84 ± 0.08) (Figure 3B). There was no significant effect of metformin and MPA combination on the invasion of USPC cells (Figure 3D).

### 3.4. MPA Reverses Tolerable Dose Metformin-Induced Inhibition of Invasion via MMP-9 and TGF-β1 in KLE Endometrial Cancer Cells

MMP-2 showed no significant changes in response to the treatments in all three cell lines (Figure 4A–C). Despite a statistical insignificance, however, MMP-9 secretion was decreased with treatment of metformin alone (0, 100 μM) and increased with combined 10μ MPA and metformin (0, 100, 1000 μM). This change of MMP-9 secretion according to various dose combinations of MPA and metformin was the same as that of cell invasion in KLE cells when treated in combination with metformin (0, 100, 1000 µM) and 10 µM MPA (3.99 ± 0.90 for control, 5.83 ± 1.04, 7.68 ± 1.38, 8.05 ± 2.09 ng/mL, respectively; *p* < 0.05) (Figure 4E). Otherwise, there were no significant changes in MMP-9 expression in Ishikawa and USPC cells (Figure 4D,F). TGF-β1 also showed similar trends to MMP-9, which was in concordance with the change in cell invasion (Figure 5B). TGF-β1 secretion was significantly decreased when KLE cells were treated with 1000 µM metformin alone compared to that in control cells (62.76 ± 2.18 vs. 54.19 ± 3.60 pg/mL; *p* = 0.024). Furthermore, TGF-β1 also exhibited the reverse pattern when treated with a combination of 1000 µM metformin and 10 µM MPA (62.76 ± 2.18 vs. 77.52 ± 5.95; *p* = 0.016). There were no significant changes in TGF-β1 levels according to the treatments in Ishikawa and USPC cells (Figure 5A,C). We showed that the reversal of 1mM metformin-induced inhibition of invasion by the treatment of 10 µM MPA was significantly inhibited again after treatment of MMP-2/9 inhibitor and/or TGF-β inhibitor. The effect of MMP-2/9 inhibitor was greater than that of TGF-β inhibitor (Figure 6). This finding implies that reversal of anti-invasive effect of metformin by the addition of MPA in KLE cells might be associated with MMP-9 and TGF-β1.

## 4. Discussion

Our study results suggest that the anticancer effects of tolerable doses of metformin varied according to the cell types and the presence of combinations with MPA in endometrial cancer. Metformin alone ≤1000 µM had anti-invasive effects on KLE cells, however, the anti-invasive effect of metformin is even reversed by the addition of 10 µM MPA. Changes in the expression of MMP-9 and TGF-β1 might be associated with these findings. We also showed that tolerable doses of metformin alone ≤1000 µM inhibited cell proliferation of Ishikawa, KLE, and USPC cells in a dose-dependent manner. However, there was no additional anti-proliferative effect of metformin ≤1000 µM and MPA co-treatment in KLE and USPC cells. 

MPA is recommended as a fertility-preserving treatment for young endometrial cancer patients, as well as palliative treatment for terminally ill patients with hormone receptor-positive cancer, especially with PR; most of the MPA anticancer effects are known to act through the interaction with PR [20]. However, response rates of MPA are unsatisfying because of the appearance of progesterone resistance; efforts were made to find an effective way to overcome this [12,20,21].

Metformin was suggested to combine with MPA based on several mechanisms of reversing progesterone resistance. Mitsuhashi et al. [22] demonstrated that the combination of MPA and metformin had significantly better prognosis than MPA alone in the patients with endometrial hyperplasia or cancer (3-year relapse-free survival, 79.3% vs. 45.2%; *p* = 0.031). And this prognosis benefit of MPA and metformin combination over MPA alone was more prominent for obese patients (BMI ≥ 25 kg/m^2^) than non-obese patients (BMI < 25 kg/m^2^) through inhibiting the phosphatidylinositol 3-kinase (PI3K)-protein kinase B (AKT)-mTOR pathway by activating AMPK, a master regulator of cellular energy homeostasis AMPK and the PI3K-AKT-mTOR pathway is known as one of the most commonly dysregulated signaling pathways in endometrial cancer [23]. One of the proposed mechanisms of action of metformin is through inhibition of complex I in the mitochondria, resulting in activation of AMPK., which then suspends ATP-consuming processes including protein synthesis, such as inhibiting mTOR. Therefore, metformin is thought as a mTOR inhibitor to reduce cellular proliferation in various type of cancer cell lines including endometrial cancer [23].

However, there were two issues to be solved. One was the supra-therapeutic concentration of metformin. The activation of AMPK was almost always demonstrated at an unrealistically high supra-therapeutic concentration of metformin, considering the maximal dose in humans (without the risk of serious side effects) [21,23,24]. An experiment of Dr. Sivalingam showed that the expressions of p-AMPKα at metformin <2 mM were weak and there was no significant decrease of cell viability at metformin <1mM in Ishikawa, HEC1A, and KLE cells [23]. A recently published randomized controlled study in human reported that short-term treatment of standard diabetic doses (850 mg per day for 3 days, and twice daily thereafter) of metformin did not reduce tumor proliferation in women with endometrial cancer awaiting hysterectomy [25]. The results did not support a biological effect of diabetic dose of metformin as well as clinical application in women with endometrial cancer. The other issue was that most of the findings were true only in Ishikawa cells, but not in other types of cells, for example, KLE [12]. Therefore, we tried to find any anti-cancer effects of tolerable doses of metformin with or without MPA in endometrial cancer cells with non-favorable clinical behavior.

We found that metformin alone at ≤1000 µM significantly inhibited the proliferation of all three cell lines (Figure 1C–E and Figure 2). The anti-proliferative effect of metformin alone at ≤1000 µM in PR-positive Ishikawa cells might be mediated through PR-B. This is because the expression of PR-B but not of p-AMPK-α increased in a dose-dependent manner with metformin treatment (Figure 2). Xie et al. [20] also demonstrated metformin significantly increased PR mRNA and protein levels in Ishikawa cells. On the other hand, growth inhibition by low-dose metformin alone of KLE and USPC cells and the enhanced anti-proliferative effect of metformin in combination with MPA in Ishikawa cells were neither associated with PR-B nor with the AMPK-dependent pathway because there were no corresponding changes in their expression levels (Figure 2). The plausible mechanism underlying this enhanced anti-proliferative effect of the metformin and MPA combination on Ishikawa cells could include AMPK-independent pathways, including factors of the Rag family of GTPases, hypoxia inducible factor (HIF) target gene, and regulated in development and the DNA damage response I (REDD1) [24]. Dr. Sivalingam also suggested that metformin might be acting through AMPK-independent pathways to inhibit mTOR in HEC1A cells based on the findings that the decreased phosphorylation of eukaryotic translation initiation factor 4E-binding protein-1 (4EBP-1) and S6, an immediate downstream target of AMPK, occurred prior to AMPK activation [23]. Other studies confirmed that there was no significant increase in p-AMPKα expression at low doses of metformin, ≤1000 µM, in Ishikawa, KLE, and USPC cells [23,26,27]. In these studies, high dose metformin ≥10 mM was shown to be necessary to bring about a significant increase in p-AMPKα expression. As there were no significant changes in PR and p-AMPKα levels in KLE and USPC cells when metformin was combined with MPA (Figure 2), we moved our focus towards invasion; the plausible invasion mechanism was not related to the activation of AMPK.

It was interesting that the dose-dependent inhibitory effect of metformin ≤1000 µM on cell invasion was found only in KLE cells, but not in Ishikawa and USPC cells. It was even more interesting that the addition of MPA to metformin resulted in the opposite effects on cell invasion in the two different types of cells, i.e., reversing metformin effect in KLE and enhancing metformin effect in Ishikawa cells (Figure 3B,C). The finding of low dose metformin alone not conferring any change in invasion capability of Ishikawa cells was consistent with that in a study of de Barros Machado et al. [28]. Even though we could not find a plausible molecular mechanism for the enhanced anti-invasive effect of the narrow dose window of metformin (0–100 µM) and MPA combination in Ishikawa cells (Figure 3B), it is notable that MMP-9 and TGF-β1 showed the same pattern of change to that of cell invasion only in KLE cells (Figure 4 and Figure 5).

Classically, MMPs are thought to facilitate cancer invasion and metastasis actively due to its ability to degrade extra-cellular matrix clearing a path for tumor cells to move through matrix barriers [29]. Among the MMPs, MMP-2 (gelatinase A), and MMP-9 (gelatinase B) are the most essential in degrading of type IV collagen, which is the main constituent of the basement membrane [30]. High expression of both MMP-2 and 9, which was observed in endometrial carcinomas, was associated with parameters of tumor aggressiveness, including advanced stage, metastasis, and lymphovascular invasion [31]. TGF-β is known to play a crucial role in the initial steps of cancer invasion associated with epithelial-mesenchymal transition (EMT) process and the acquisition of an invasive/migratory phenotype during myometrial infiltration and metastasis in endometrial cancer [30,32,33]. Therefore, we decided to look at MMP-2, MMP-9 and TGF-β for the plausible underlying mechanisms for the anti-invasive effect of metformin as well as its reversal effect in the addition of MPA [19,31,32,34].

Samarnthai et al. [35] reported the dualistic model of endometrial carcinoma, type I and type II, in terms of genetic changes and clinical behavior. KLE cells could be clinically characterized as type II cancer cells because of the aggressive behavior and poor outcomes, but also as type I due to the frequent PTEN and KRAS mutations and rare p53 mutation, which are typical in type I cancer. Zhang et al. showed that the expression of glyoxalase I (GloI) was significantly higher in KLE cells than Ishikawa cells, suggesting GloI might be involved in progestin resistance in KLE cells [12]. They reported metformin could reverse progestin resistance by downregulating GloI expression. Regarding cell invasion and migration, Wen et al. demonstrated that suppression of golgi phosphoprotein 3 (GOLPH3), a novel oncogene, of which the expression was the highest among four endometrial cancer cell lines (HEC1A, KLE, RL95-2, and Ishikawa), using shGOLPH3 could reduce KLE cell proliferation, migration, and invasion while accelerating apoptosis [36]. Furthermore, they found that the number of nude mice with distant metastasis was smaller in KLE-shGOLPH3 injection mice group (4/7) than that in control KLE injection group (7/7).

Despite a small number of studies, so far, addressing the anti-invasive and/or anti-migratory effects of metformin in endometrial cancer cells, this study is, to the best of our knowledge, the first study which showed that the significant anti-invasive effect of a tolerable dose of metformin in KLE cells was completely reversed to the state of no treatment by the addition of MPA; these findings might be mediated through MMP-9 and TGF-β1. However, our study has some limitations firstly in that metformin was not shown as an AMPK activator. There are a few studies which did not support metformin as a potent AMPK activator, not only in proliferation [23,26], but also in invasion [19]. Secondly, given the continuously decreasing trend of KLE cell invasion with the increase of metformin dose (0, 100, 1000 μM), MMP-9 concentration at 1000 µM metformin only was expected to be lower than that of 100 µM metformin. Thirdly, the effects of MMP-2/9 or TGF-β inhibitor alone on cell invasion were not shown (Figure 6). It could weaken our conclusion on the molecular mechanism of reversal of anti-invasive effect of metformin by the addition of MPA in KLE cells. Lastly, study results only from in vitro experiments without supportive in vivo animal study kept us from drawing a firm conclusion.

Most studies on metformin and MPA in endometrial cancer have concluded that combining the two could be a potential therapeutic strategy for overcoming progesterone resistance [12,20]. However, our study suggests the possibility of the combination being harmful instead of beneficial in some conditions, especially in clinically highly aggressive cancers but genetically classified as type I. Further animal studies are required to clinically confirm our study findings.

## Figures and Tables

**Figure 1 jcm-09-03585-f001:**
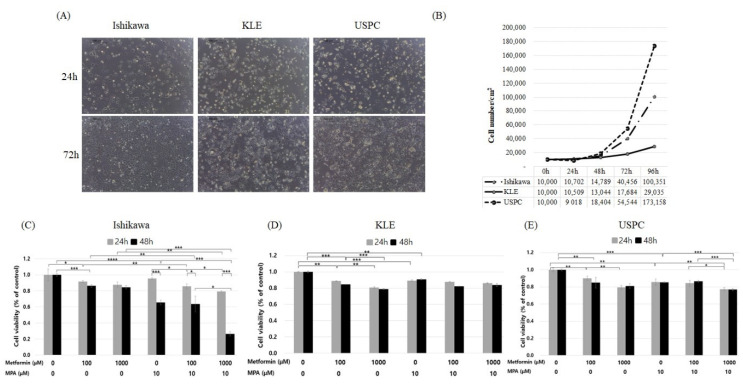
Cell growth and growth inhibition by metformin and medroxyprogesterone (MPA) in three endometrial cancer cell lines: Ishikawa, KLE, and USPC. (**A**) Cell morphology and number at 24 h and 72 h, (**B**) cell growth rate (0, 24, 48, 72, and 96 h), cell viability after treatment of different dose combinations of MPA (0, 10 µM) and metformin (0, 100, 1000 µM) in Ishikawa (**C**), KLE (**D**), and USPC cells (**E**). Each experiment was independently performed in triplicate: results are expressed as mean ± standard deviation (**C**–**E**). Multiple testing correction was performed using Bonferroni adjustment and Bonferroni corrected p-values were used for statistical significance (**C**–**E**). * *p* < 0.05, ** *p* < 0.005, *** *p* < 0.0005, **** *p* < 0.00005.

**Figure 2 jcm-09-03585-f002:**
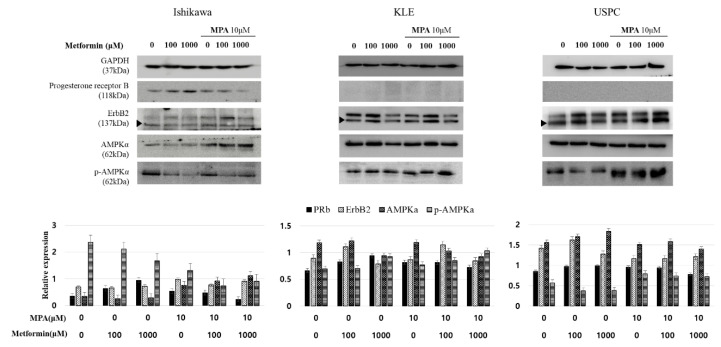
Expression of progesterone receptor B, ErbB2, AMPKα, and phospho-AMPKα in three endometrial cancer cell lines (Ishikawa, KLE, and USPC) according to treatment with different dose combinations of medroxyprogesterone (MPA) (0, 10 µM) and metformin (0, 100, 1000 µM). Band densities are quantified using ImageJ and values are presented as mean ± S.E. of independent triplicates per group. Progesterone receptor B, ErbB2, and AMPKα were normalized to GAPDH, whilst p-AMPKα was normalized to AMPKα.

**Figure 3 jcm-09-03585-f003:**
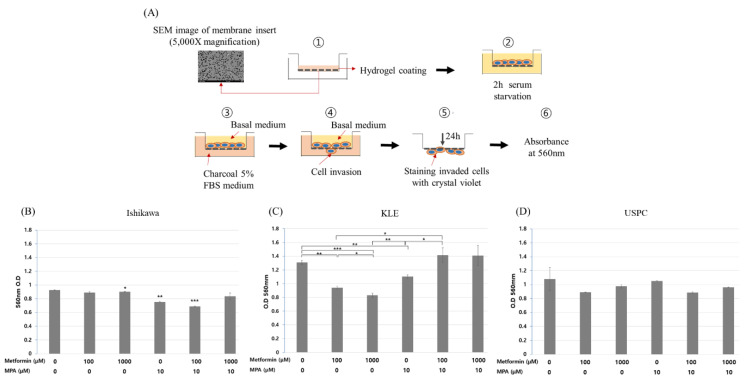
Invasion assay in three endometrial cancer cell lines (Ishikawa, KLE, and USPC). (**A**) The process of invasion assay, cell invasion after treatment of different dose combinations of medroxyprogesterone (MPA) (0, 10 µM) and metformin (0, 100, 1000 µM) in Ishikawa (**B**), KLE (**C**), and USPC cells (**D**). (**B**) All comparisons were with control (no metformin/MPA). (**B**–**D**) Each experiment was independently performed in triplicate: results are expressed as mean ± standard deviation. Multiple testing correction was performed using Bonferroni adjustment and Bonferroni corrected p-values were used for statistical significance (**B**–**D**). * *p* < 0.05, ** *p* < 0.005, *** *p* < 0.0005.

**Figure 4 jcm-09-03585-f004:**
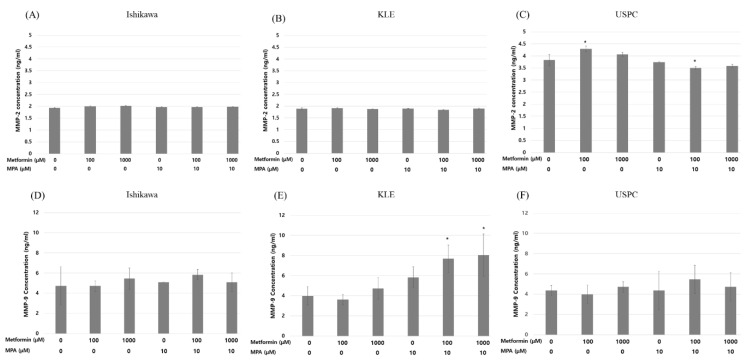
Matrix metallopeptidase (MMP)-2 (**A**, Ishikawa; **B**, KLE; **C**, USPC) and MMP-9 (**D**, Ishikawa; **E**, KLE, **F**, USPC) in three endometrial cancer cell lines after treatment with different dose combinations of medroxyprogesterone (MPA) (0, 10 µM) and metformin (0, 100, 1000 µM). Each experiment was independently performed in triplicate: results are expressed as mean ± standard deviation. Multiple testing correction was performed using Bonferroni adjustment and Bonferroni corrected p-values were used for statistical significance. * *p* < 0.05.

**Figure 5 jcm-09-03585-f005:**
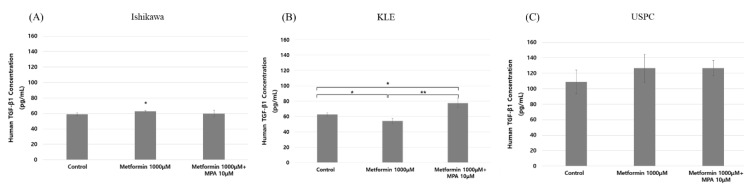
Transforming growth factor (TGF)-β1 in three endometrial cancer cell lines after treatment with different dose combinations of medroxyprogesterone (MPA) (0, 10 µM) and metformin (0, 1000 µM) in Ishikawa (**A**), KLE (**B**), and USPC cells (**C**). Each experiment was independently performed in triplicate: results are expressed as mean ± standard deviation. Multiple testing correction was performed using Bonferroni adjustment and Bonferroni corrected *p*-values were used for statistical significance. * *p* < 0.05, ** *p* < 0.005.

**Figure 6 jcm-09-03585-f006:**
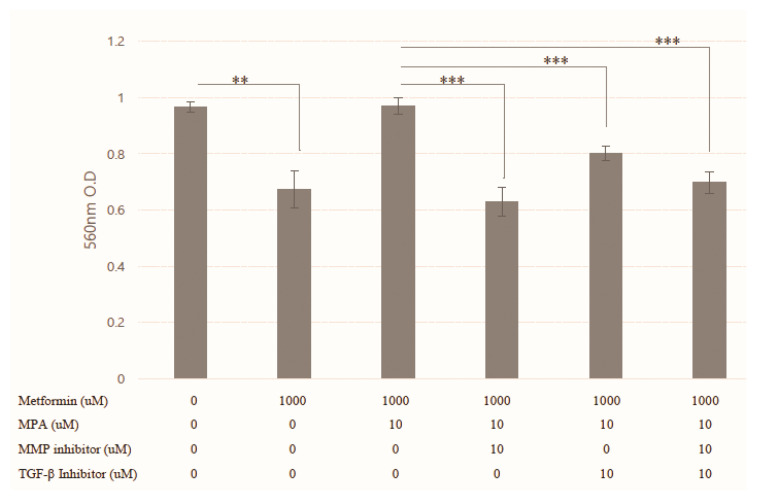
Cell invasion after combination treatment of 1mM metformin (Met), 10 µM medroxyprogesterone (MPA), 10 µM Matrix metallopeptidase (MMP) inhibitor and transforming growth factor (TGF)-β inhibitor in KLE cell lines. Each experiment was independently performed in triplicate: results are expressed as mean ± standard deviation. Multiple testing correction was performed using Bonferroni adjustment and Bonferroni corrected *p*-values were used for statistical significance. ** *p* < 0.005, *** *p* < 0.0005.

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
