# Peer review of "Medroxyprogesterone Reverses Tolerable Dose Metformin-Induced Inhibition of Invasion via Matrix Metallopeptidase-9 and Transforming Growth Factor-β1 in KLE Endometrial Cancer Cells"

_jcm, 2020, doi:10.3390/jcm9113585_

Round 1

Reviewer 1 Report

Very good study to understand the anticancer effect of tolerable doses of metformin alone or with MPA in endometrial cancer cells. Here authors are showing the possibility of the combination therapy (MPA and Metformin) can be harmful instead of beneficial in some conditions, especially in clinically highly aggressive but genetically type I cancer. There are few suggestions which can improve the quality of manuscript:

  1. All figures need to have clear resolution. It is not readable.
  2. In figure no.2 ErbB2 IB pic, authors need to change.
  3. Authors need to confirm this study in in-vivo too.

Author Response

  1. All figures need to have clear resolution. It is not readable.
  • Response: all figures are updated so that those are clearly understandable.  
  1. In figure no.2 ErbB2 IB pic, authors need to change.
  • Response: ErbB2 IB picture was changed as you recommended. Thank you.
  1. Authors need to confirm this study in in-vivo too.
  • Response: Thank you for your suggestion. We totally agree with you. Unfortunately, in vivo study data is not available for now. We also failed to find any reference in vivo studies which can confirm our study results probably because most of relevant in vivo studies primarily focused on tumor growth instead of invasion and/or migration. Nevertheless, based on the originality of our in vitro study results, we are planning an in vivo animal study. Thank you again for your valuable comments.

Reviewer 2 Report

In this manuscript, Suh et al. investigated the effects of metformin +/- MPA on cell growth, viability and invasion in three endometrial cancer cell lines. Combined metformin + MPA only substantially reduced viability in Ishikawa cells, whilst invasion was not substantially reduced in any cell line. Metformin alone reduced invasion in KLE cells, which was inhibited by the addition of MPA. The authors attempted to explain the mechanism for how MPA prevented the inhibitory effects of metformin in KLE cells by focusing on MMPs and TGF-b1.

The authors need to explain the rationale for looking at MMP-2, MMP-9 and TGFb secretion in this manuscript. Methods section is missing some details. Statistical analyses need to be improved. Text in figure legends is very small and difficult to read, please use a larger font size. Also state what error bars are (SD, SEM?) and how many replicates (technical and/or biological) were done. Conclusions are over-stated and need to be justified. A discussion of human studies exploring metformin +/- MPA would be useful. E.g. A Human study in women with atypical hyperplasia or endometrioid endometrial cancer who were given pre-surgical metformin or placebo showed that post-treatment Ki-67 expression was similar between both arms, indicating that metformin does not decrease tumour proliferation (Kitson et al., 2019). Women undergoing fertility-sparing treatment have been successfully treated with MPA + metformin. Mitsuhashi et al. (2019) showed metformin may be more efficacious in women with BMI>25kg/m2. English needs some correcting.

Specific comments:

Abstract Lines 26-28: Please re-word these sentences as their meaning is currently unclear.

Line 69: Specify the name of the USPC cell line.

Line 89: Presumably “increase” should be “measure”?

Line 99: What is the rationale for using 10uM MPA?

Cell growth analysis was done at 24-hour intervals until 96 hours, however, cell survival analysis was only done until 48 hours – why the discrepancy in timings between experiments?

Lines 107-108: Add details for GAPDH antibody.

Lines 118-119 and 125-126: Please clearly explain what “treated according to the conditions at the time of the exchange of the medium” means. What were the concentrations of each inhibitor used, when were they applied to the cells and how long were they applied for?

Lines 131-132: Student’s t-test can only be applied to normally distributed data – are the data normally distributed (e.g. use Shapiro-Wilk test)? If not, then data should be log-transformed or a different test used, e.g. Mann-Whitney U-test. Correction also needs to be applied for multiple testing and methodology for doing this stated.

Line 151: States AMPK levels are changed, this should be p-AMPK.

Line 161: Figure 2 shows that p-AMPKa is increased with MPA alone and in combination with metformin in USPC cells, the following sentence also states this, yet this sentence states that p-AMPK is not changed in USPC – please correct.

Lines 163-165: What does the sentence “However, there was neither dose dependency in expression patterns nor consistency with anti-proliferative effects” mean?

Lines 170-172: Suggest deleting this sentence.

Lines 178-179: Metformin or MPA alone reduce invasion in KLE cells, but when combined, invasion is similar to control (blank media), it is not ‘stimulated’.

Lines 180-184: If the authors want to claim that MPA + metformin is synergistic in Ishikawa cells, then they must do the experiments to show this.

Lines 194-197: I don’t understand this sentence - are the authors trying to state that MMP-9 secretion is increased with combined MPA + metformin and that invasion is also increased with combined MPA + metformin (Figure 3C)? Please explain the rationale for linking MMP-9 and invasion.

Line 206: Please note that “re-inhibited” is not a word.

Lines 208-209: The effects of MMP or TGF-b inhibitor alone on invasion need to be shown before any conclusions can be drawn. How specific are the inhibitors, are there any known off-target effects? How were the concentrations of either inhibitor selected? Ideally, function would be assessed by looking at the activity of downstream markers. E.g. SMAD phosphorylation is downstream of TGF-b, is this reduced with 10uM TGF-b inhibitor?

Lines 225-226: What is the rationale for the statement that “Changes in the expression of MMP-9 and TGF-β1 were plausible mechanisms underlying these findings”?

Lines 228-229, 249, 251-252, 268: The authors have not properly investigated additive or synergistic effects of metformin and MPA in this manuscript. If the authors want to draw these conclusions then they need to perform the required experiments and analyses.

Lines 245-250: The conclusion that metformin acts through PR is not justified – metformin also reduced viability in KLE and USPC (Figures 1D-E) cells which are PR-negative (Figure 2). Several publications have shown that PR expression is increased with Metformin, these should be referenced.

Line 261-262: What is known about KLE cells that could potentially explain their sensitivity to metformin vs Ishikawa and USPC cells? There is some discussion in lines 279-283 but this could be more detailed.

Lines 271-272: Previous studies have shown that MMP-2 and -9 expression are affected by metformin.

Lines 276-279: If the authors want to draw this conclusion then they need to show evidence to support it or reference relevant literature.

Lines 291-293: Unclear what the authors are trying to state here.

Line 298: A limitation of this study is that it has only been conducted in vitro. A brief discussion of what human studies to date have shown would be useful.  

Figure 1B: Can the authors please explain why the USPC cell number has decreased at 24hrs.

Figures 1C-E: What do the grey and black bars represent – are these different time points? The authors have performed a multitude of statistical tests, have these been corrected for multiple testing? If not, then this must be done.

Figure 1C: Ishikawa + 100uM Metformin vs Ishikawa + 1000uM Metformin is p<0.0005, yet the data look similar, could the authors please explain this.

Figure 1C: MPA + 100uM Metformin (black bars) appears to have little effect compared to MPA alone, decreased viability is only seen with MPA + 1000uM metformin – please ensure the description of results in the text is accurate (lines 142-143) and insert more detailed explanation of results.

Figure 2: GAPDH exposure is very long and does not appear to be equal between samples. Could the authors please provide a shorter exposure of GAPDH to assess loading between samples – if the loading control is not equal then this experiment needs to be repeated.

Figure 2: Total AMPKa expression is not equal between samples in Ishikawa cells – is this because loading is unequal (see comment about GAPDH) or could the authors please explain why expression would be altered by metformin and/or MPA.

Figure 2: Could the authors please explain why ErbB2 levels are shown and their relevance to this manuscript? Also, what is the rationale for looking at progesterone receptor B (PR-B) as opposed to, or in combination with, total PR?

Figure 2: Please add band quantification (e.g. use ImageJ).

Figure 3B-D: The authors need to explain what comparisons are being tested here – are all conditions being compared to control (no metformin/MPA)? Has multiple testing been applied? If not, then this needs to be done. What do the error bars represent?

Figure 3B: 1000uM metformin looks similar to 100uM metformin, yet 1000uM is significant (p<0.05) yet 100uM is not – please explain.

Figure 4C: MMP-2 secretion from USPC cells is much higher than from Ishikawa or KLE cells – do the authors have an explanation for this?

Figure 5: Why was TGF-b1 only assessed with 1000uM metformin +/- 10uM? For consistency, the authors should show all data as per other figures.

Figure 5C: As for MMP-2, TGF-b1 secretion from USPC cells is much higher than from Ishikawa or KLE cells – do the authors have an explanation for this?

Figure 6: The authors need to show the effects of MMP or TGF-b inhibitor alone on invasion.

Author Response

Reviewer 2

  1. The authors need to explain the rationale for looking at MMP-2, MMP-9 and TGFb secretion in this manuscript.
    • Response: thank you for your comments. We explained the rationale for looking at MMP-2, MMP-9 and TGFb secretion at discussion section as you recommended.

Methods section is missing some details. Statistical analyses need to be improved.

  • Response: thank you for your comments. Methods section was revised with more details.

Regarding statistical analyses, the Shapiro-Wilk test was used to check if the data is normally distributed. Test results confirmed all data were normally distributed (p value <0.05). Therefore, we kept using student t-test to compare mean of two groups. We added this on statistical analysis of methods section.

Text in figure legends is very small and difficult to read, please use a larger font size.

  • Response: I am sorry for this point which might cause inconvenience to reviewers. Texts in all figures were updated and became big enough to read. Thank you for your comments.

Also state what error bars are (SD, SEM)

  • Response: error bars stand for standard deviation. We added relevant explanation at figure lagend of every figure if needed.

and how many replicates (technical and/or biological) were done.

  • Response: basically, all experiments were performed in triplicate. We described this at figure legends. Thank you for your comments.

Conclusions are over-stated and need to be justified.

  • Response: the last conclusion, that is, this might be mediated through MMP-9 and TGF-β1, could be thought to be over-stated and need to be justified.

A discussion of human studies exploring metformin +/- MPA would be useful. E.g. A Human study in women with atypical hyperplasia or endometrioid endometrial cancer who were given pre-surgical metformin or placebo showed that post-treatment Ki-67 expression was similar between both arms, indicating that metformin does not decrease tumour proliferation (Kitson et al., 2019). Women undergoing fertility-sparing treatment have been successfully treated with MPA + metformin. Mitsuhashi et al. (2019) showed metformin may be more efficacious in women with BMI>25kg/m2.

  • Response: thank you for your comments. We added the references of human studies in appropriate part in discussion, which definitely made the discussion section more comprehensive and supporting than before. Thank you again.

English needs some correcting

  • Response: English was edited before submission of revised manuscript. Thank you.

Specific comments:

Abstract Lines 26-28: Please re-word these sentences as their meaning is currently unclear.

  • Response: the sentences were re-worded as following: was combined, which was significantly inhibited again after treatment of MMP-2/9 inhibitor and/or TGF-β inhibitor. Changes of MMP-9 and TGF-β1 according to combinations of MPA and metformin were similar to those of invasion in KLE cells.

Line 69: Specify the name of the USPC cell line.

  • Response: the name of USPC cell line was USPC-ARK-1. It was purchased from Dr. Alessandro Santin of Yale University, not from ATCC. We made a correction on methods section accordingly.

Line 89: Presumably “increase” should be “measure”?

  • Response: “increase” was replaced with “measure”. Thank you for your comments.

Line 99: What is the rationale for using 10uM MPA?

  • Response: we added the rationale for using 10uM MPA at methods section. Thank you for your comments.

MPA dose of 10uM was set based on a study of Zhang et al. [12] which also evaluated the anticancer effect of MPA and metformin combination in endometrial cancer cells. MPA 10uM was high enough to inhibit proliferation of Ishikawa cells. However, it was too low to suppress progestin-resistant Ishikawa cells, which were considered to have similar characteristics with KLE cells in our study. Progestin resistance of progestin-resistant Ishikawa cells was overcome by the addition of metformin to MPA 10uM [12]. KLE and USPC cells were not expected to be susceptible to higher dose of MPA because of negative expression of ER and PR. Therefore, 10uM was set as an optimal dose of MPA which could show possible anticancer effects when combined with metformin in these cell lines. There was another study of progestin and metformin in endometrial cancer showed that 10uM MPA was minimal dose that could significantly inhibit growth of RL95-2 cells (ER+/PR+) [15].

Cell growth analysis was done at 24-hour intervals until 96 hours, however, cell survival analysis was only done until 48 hours – why the discrepancy in timings between experiments?

  • Response: cell growth analysis without any intervention usually keeps counting cell number per cm2 regularly until the time of next passage culture, which is around 72~96 hrs. We evaluated cell viability in cell survival analysis three times at 0, 24 and 48 hr because three times are usually enough to find any trend of change. Thank you for your comments.       

Lines 107-108: Add details for GAPDH antibody.

  • Response: details of GAPDH antibody were added at methods section. Thank you.

Lines 118-119 and 125-126: Please clearly explain what “treated according to the conditions at the time of the exchange of the medium” means. What were the concentrations of each inhibitor used, when were they applied to the cells and how long were they applied for?

  • Response: thank you your comments. We admit those expressions were not easy to understand and not fully informative. We made corrections on those sentences with addition of essential details you requested. Thank you again.

Lines 131-132: Student’s t-test can only be applied to normally distributed data – are the data normally distributed (e.g. use Shapiro-Wilk test)? If not, then data should be log-transformed or a different test used, e.g. Mann-Whitney U-test.

  • Response: thank you for your comments. As you recommended, Shapiro-Wilk test was used to check if the data were normally distributed. Test results confirmed all data were normally distributed (p value > 0.05). Therefore, we kept using student t-test to compare means of two groups. We added this on statistical analysis of methods section.

Correction also needs to be applied for multiple testing and methodology for doing this stated.

  • Response: thank you for your comments. Each experiment was independently performed in triplicate: results are expressed as mean ± standard deviation. We added this sentence to figure legend if appropriate.

Line 151: States AMPK levels are changed, this should be p-AMPK.

  • Response: thank you for your comment. We changed AMPK to p-AMPKα.

Line 161: Figure 2 shows that p-AMPKa is increased with MPA alone and in combination with metformin in USPC cells, the following sentence also states this, yet this sentence states that p-AMPK is not changed in USPC – please correct.

  • Response: thank you for your comments. We deleted p-AMPKα from the sentence.

Lines 163-165: What does the sentence “However, there was neither dose dependency in expression patterns nor consistency with anti-proliferative effects” mean?

  • Response: thank you for your comments. We admit this sentence is unclear. The sentence was changed as follows: In AMPK/mTOR pathway, high expression of p-AMPKα results in growth inhibition via inhibiting mTOR. Even though the expression of p-AMPKα was stronger when both metformin and MPA were used than when metformin was used alone in USPC cells (Figure 2), cell growths of the combination group were not significantly lower than those of metformin alone group (Figure 1E). In addition, there was no dose-dependent increase of p-AMPKα expression according to metformin doses (0, 100, 1000 µM) in USPC cells (Figure 1E).

Lines 170-172: Suggest deleting this sentence.

  • Response: as you recommended, the sentence was deleted. Thank you for your comments.

Lines 178-179: Metformin or MPA alone reduce invasion in KLE cells, but when combined, invasion is similar to control (blank media), it is not ‘stimulated’.

  • Response: thank you for your comment. We agree with you and revised all relevant parts throughout our manuscript with deleting “stimulating”.

Lines 180-184: If the authors want to claim that MPA + metformin is synergistic in Ishikawa cells, then they must do the experiments to show this.

  • Response: thank you for your comments. We agree with you. Instead of doing additional experiments to prove any synergism between MPA and metformin in inhibiting cell invasion, we rephrased relevant sentences with deleting “synergistic effect”.

Lines 194-197: I don’t understand this sentence - are the authors trying to state that MMP-9 secretion is increased with combined MPA + metformin and that invasion is also increased with combined MPA + metformin (Figure 3C)?

  • Response: thank you for your comment. We reworded this sentence to make it more clearly.

Despite a statistical insignificance, however, MMP-9 secretion was decreased with treatment of metformin alone (0, 100 μM) and increased with combined 10μ MPA and metformin (0, 100, 1000 μM). This change of MMP-9 secretion according to various dose combinations of MPA and metformin was the same as that of cell invasion in KLE cells when treated in combination with metformin (0, 100, 1000 µM) and 10 µM MPA.

Please explain the rationale for linking MMP-9 and invasion.

  • Response: we added the rationale for linking MMP-9 and invasion in discussion section. Thank you for your comment.

Line 206: Please note that “re-inhibited” is not a word.

  • Response: we changed the expression using “inhibited again” instead of “re-inhibited”. Thank you for your comment.

Lines 208-209: The effects of MMP or TGF-b inhibitor alone on invasion need to be shown before any conclusions can be drawn.

  • Response: thank you for your comment. Yes, we agree with you that proving the effects of MMP or TGF-β inhibitor alone on cell invasion could lead us to be able to draw a firm conclusion. However, additional data of the effects of MMP or TGF-β inhibitor alone on cell invasion is not available for now. We added this point as a limitation of our study at discussion section.

How specific are the inhibitors, are there any known off-target effects? How were the concentrations of either inhibitor selected?

  • Response: thank you for your comments.

As you mentioned, lack of selectivity and subsequent off-target effects of MMP inhibitors have been big problems in developing anticancer drugs using MMP inhibitors (Overall et al. British Journal of Cancer. 2006). However, MMP-2/MMP-9 inhibitor I-CAS 193807-58-8-Calbiochem (Sigma-Aldrich Korea) we used in our experiment was shown to have highly selective in vitro and in vivo activities because of its structure, a series of aryl sulfonamide derivatives containing biaryl, tetrazole, amide, and triple bond (Tamura et al. Journal of medicinal chemistry 1998). We don’t think there were significant off-target effects in our experiments.

Regarding concentrations of inhibitors, relevant references were added at methods section.  

Ideally, function would be assessed by looking at the activity of downstream markers. E.g. SMAD phosphorylation is downstream of TGF-b, is this reduced with 10uM TGF-b inhibitor?

  • Response: thank you for your comment. Unfortunately, we did not look at SMAD phosphorylation as a downstream molecule of TGF-β at this time. We are planning to study in-depth mechanism of action of anti-invasive effects of metformin and MPA in endometrial cancer in vitro and in vivo as a next step.

Lines 225-226: What is the rationale for the statement that “Changes in the expression of MMP-9 and TGF-β1 were plausible mechanisms underlying these findings”?

  • Response: thank you for your comment. I understand why you questioned on this point. I admit we did not conduct a perfect mechanism-of-action study to assess any change of the activity of molecules in a downstream pathway. Nevertheless, we found that MMP-9 secretion, but, not MMP-2 secretion, was decreased with treatment of metformin alone (0, 100 μM) and increased with combined 10μ MPA and metformin (0, 100, 1000 μM) which was the same as the changes of cell invasion in KLE cells when treated in combination with metformin (0, 100, 1000 µM) and 10 µM MPA. Moreover, we showed that the reversal of 1mM metformin-induced inhibition of invasion by the treatment of 10 µM MPA was significantly inhibited again after treatment of MMP-2/9 inhibitor and/or TGF-β The effect of MMP-2/9 inhibitor was greater than that of TGF-β inhibitor. Based on these results, we stated that changes in the expression of MMP-9 and TGF-β1 were plausible mechanisms underlying these findings.

Considering the limitation of the current study, we tried to toned down the conclusion.    

Lines 228-229, 249, 251-252, 268: The authors have not properly investigated additive or synergistic effects of metformin and MPA in this manuscript. If the authors want to draw these conclusions then they need to perform the required experiments and analyses.

  • Response: thank you for your comments. We reworded relevant sentences using other expression instead of “synergistic effect”.

Lines 245-250: The conclusion that metformin acts through PR is not justified – metformin also reduced viability in KLE and USPC (Figures 1D-E) cells which are PR-negative (Figure 2). Several publications have shown that PR expression is increased with Metformin, these should be referenced.

  • Response: thank you for your comments. However, action of metformin through PR was only for PR-positive Ishikawa cells, which was already described in this way in the manuscript. We added a reference which showed that PR expression was increased with metformin.  

Line 261-262: What is known about KLE cells that could potentially explain their sensitivity to metformin vs Ishikawa and USPC cells? There is some discussion in lines 279-283 but this could be more detailed.

  • Response: thank you for your comments. As you mentioned, we briefly described the moleculogenetic and clinical characteristics of KLE cells in discussion. Because of its complexity, type of KLE cells is inconsistent in previous literature. Some classified KLE as type I and others thought it as type II. We added more details on KLE cells with referencing relevant studies.

Lines 271-272: Previous studies have shown that MMP-2 and -9 expression are affected by metformin.

  • Response: we deleted this sentence and rephrased. Thank you for your comment.

Lines 276-279: If the authors want to draw this conclusion then they need to show evidence to support it or reference relevant literature.

  • Response: thank you for your comments. We deleted the sentence because we cannot find relevant literature for reference.

Lines 291-293: Unclear what the authors are trying to state here.

  • Response: we reworded that sentence to make it clear. Thank you for your comment.

Line 298: A limitation of this study is that it has only been conducted in vitro. A brief discussion of what human studies to date have shown would be useful.  

  • Response: thank you for your comment. The combined treatment of MPA and metformin could do harm rather than doing good for patients with poorly differentiated type I endometrial cancer. Therefore, in vivo animal study should precede human studies. Given this condition, we added a few relevant human studies in discussion section.

Figure 1B: Can the authors please explain why the USPC cell number has decreased at 24hrs.

  • Response: thank you for your comment. The graphs show that USPC cells had the fastest growth among the three cell lines. However, cell growth of three cell lines did not look quite different until 48 hours. Although the small decrease of USPC cell number is considered insignificant, the possible causes includes: taking more time to attach, not entering active phase of proliferation, and pipetting error.

Figures 1C-E: What do the grey and black bars represent – are these different time points? The authors have performed a multitude of statistical tests, have these been corrected for multiple testing? If not, then this must be done.

  • Response: thank you for your comments. Grey and black bars represent 24hours and 48hours, respectively. Legend is too small to be seen in the original figure. We revised all figures with using larger font size. As for statistical tests, we performed every experiment three times independently. As you recommended at the former comment, Shapiro-Wilk test was used to check the distribution of data and the test results confirmed that all data were normally distributed (p value > 0.05). Therefore, we kept using student t-test to compare means of two groups. We added this on statistical analysis of methods section.  

Figure 1C: Ishikawa + 100uM Metformin vs Ishikawa + 1000uM Metformin is p<0.0005, yet the data look similar, could the authors please explain this.

  • Response: thank you for your comment. There was a mistake at this point. The comparison you mentioned was statistically insignificant with p-value >0.05. We removed a corresponding marker from the figure 1C.

Figure 1C: MPA + 100uM Metformin (black bars) appears to have little effect compared to MPA alone, decreased viability is only seen with MPA + 1000uM metformin – please ensure the description of results in the text is accurate (lines 142-143) and insert more detailed explanation of results.

  • Response: thank you for your comment. Difference between the two values were small. However, linear regression analysis showed the trend of decreasing was significant with p-value<0.05. Therefore, we can ensure the current description is accurate. We updated statistical analysis in methods section accordingly. Thank you again.

Figure 2: GAPDH exposure is very long and does not appear to be equal between samples. Could the authors please provide a shorter exposure of GAPDH to assess loading between samples – if the loading control is not equal then this experiment needs to be repeated.

  • Response: thank you for your comment. We admit that GAPDH exposure was very long and does not appear to be equal between samples. We revised WB pictures.

Figure 2: Total AMPKa expression is not equal between samples in Ishikawa cells – is this because loading is unequal (see comment about GAPDH) or could the authors please explain why expression would be altered by metformin and/or MPA.

  • Response: thank you for your comment. We also updated WB pictures of total AMPKα.

Figure 2: Could the authors please explain why ErbB2 levels are shown and their relevance to this manuscript? Also, what is the rationale for looking at progesterone receptor B (PR-B) as opposed to, or in combination with, total PR?

  • Response: thank you for your comments. Overexpression of receptor tyrosine kinase ErbB2(HER2), a member of EGFR, was found in 3-10% of type I endometrial cancers and in max. 30% of type II serous carcinoma. (Dedes et al. Nat Rev Clin Oncol 2011;8:261-71) USPC cells represent type II serous carcinoma. Strong expression of ErbB2 in USPC cells is associated with cell invasion and migration suggesting poor prognosis of serous carcinoma. HER2 overexpression was shown to have inverse correlation with hormone receptor, including PR. Metformin has been shown to suppress both the tyrosine kinase activity and the expression of the HER2 in in vitromodels of HER2-overexpressing breast cancer cells (Martin-Castillo et al. Oncotarget 2018;9(86):35687-704).

Lower levels of PR isotype B (PR‐B) have been associated with compromised survival, suggesting a critical role for PR‐B in the development of endometrial cancer. EGFR expression was higher in progestin-resistant KLE cells than in progestin-sensitive Ishikawa cells. In contrast, PR-B expression was higher in Ishikawa cells than KLE cells. Higher EGFR expression reduced sensitivity to progestin and decreased PR-B expression in Ishikawa cells. (Ai et al. Cancer 2010 Aug;116(15):3603-13) Taken together, PR-B (rather than total PR) and ErbB2 were chosen as a marker of sensitivity to progestin and metformin, respectively.

Figure 2: Please add band quantification (e.g. use ImageJ).

  • Response: thank you for your comment. We added band quantification using ImageJ.

Figure 3B-D: The authors need to explain what comparisons are being tested here – are all conditions being compared to control (no metformin/MPA)? Has multiple testing been applied? If not, then this needs to be done. What do the error bars represent?

  • Response: thank you for your comment. We described all requested information at figure legend.

Figure 3B: 1000uM metformin looks similar to 100uM metformin, yet 1000uM is significant (p<0.05) yet 100uM is not – please explain.

  • Response: thank you for your comment. Yes, it is true. Mean±SD of control, 100uM metformin, and 1000uM metformin were 0.9263±0.0055, 0.890±0.0220, and 0.9020±0.0090. For the comparison of control and 100uM metformin, t=2.749 and p=0.051. By contrast, t=3.994 and p=0.016 for the comparison between control and 1000uM metformin. This might be caused by large SD of 100uM metformin.

Figure 4C: MMP-2 secretion from USPC cells is much higher than from Ishikawa or KLE cells – do the authors have an explanation for this?

  • Response: USPC cells have a typical characteristic of type II endometrial cancer. Type II cancer has a strong tendency for metastasis compared to type I cancer. MMP-2 is the most essential in degrading of type IV collagen of basement membrane and EMT. There are many literatures which support high expression of MMP-2 in type II cancer including USPC. On the contrary, KLE and Ishikawa cells originally belong to type I cancer although KLE shares aggressive clinical behavior with type II cancer. Therefore, we speculate that high expression of MMP-2 in USPC compared with Ishikawa and KLE because of its inherent characteristics of favoring EMT.  

Figure 5: Why was TGF-b1 only assessed with 1000uM metformin +/- 10uM? For consistency, the authors should show all data as per other figures.

  • Response: thank you for your comment. I agree with you. Unfortunately, however, the data of other dose combinations of MPA and metformin is not available for now.

Figure 5C: As for MMP-2, TGF-b1 secretion from USPC cells is much higher than from Ishikawa or KLE cells – do the authors have an explanation for this?

  • Response: thank you for your comment. Similar to MMP-2, TGFβ1 is known to play a crucial role in the initial steps of cancer invasion associated with EMT and the acquisition of an invasive/migratory phenotype during metastasis in cancer. Therefore, TGF-b1 secretion from USPC cells is supposed to be much higher than from Ishikawa or KLE cells.

Figure 6: The authors need to show the effects of MMP or TGF-b inhibitor alone on invasion.

  • Response: thank you for your comment. As we response to your comment on line 208-209, additional data of the effects of MMP or TGF-β inhibitor alone on cell invasion is not available for now. We added this point as a limitation of our study at discussion section.

Reviewer 3 Report

I carefully read and evaluated the paper “Medroxyprogesterone reverses tolerable dose metformin-induced inhibition of invasion via matrix metallopeptidase-9 and transforming growth factor-β1 in KLE endometrial cancer cells”.

The aim of the authors is to evaluate the anticancer effect of tolerable doses of metformin with or without medroxyprogesterone (MPA) in different cell lines of endometrial cancer.

The authors have analysed cell viability, cell invasion, and levels of matrix metallopeptidaseand transforming growth factor TGF)-β1 in three human endometrial adenocarcinoma cell lines (Ishikawa, KLE, and USPC) after treatment with different dose combinations of metformin and MPA.

I have the following considerations:

  • What is the purpose of the study? That of the discussion seems different from that of the introduction. Clarify it.
  • Can the authors better explain how they selected the increasing doses of metformin? How they select the maximum dose?
  • Can the authors better explain the AMPKalfa pathway and its function in cell proliferation?
  • Can the authors better explain how the evaluate cell proliferation?
  • Could the decrease in progesterone receptors induce resistance to MPA?

Author Response

What is the purpose of the study? That of the discussion seems different from that of the introduction. Clarify it.

  • Response: thank you for your comment. We admit the purpose was described unclearly and inconsistently throughout the manuscript. The purpose of this study was to evaluate the anticancer effect of tolerable doses of metformin with or without MPA as well as its underlying molecular mechanisms in endometrial cancer cells. We changed relevant parts in the manuscript. Thank you again.

Can the authors better explain how they selected the increasing doses of metformin? How they select the maximum dose?

  • Response: thank you for your comment. We updated relevant methods section with more details as follows:

The tolerable doses of metformin which could achieve a plasma concentration of around 1 mg/L was 500 mg twice/day [14]. Although the maximal approved total daily dose of metformin for treatment of diabetes mellitus is 2.5 g (35 mg/kg body weight)[8], slow but progressive increase of fasting lactic acid levels were noted during metformin treatment with multiple doses from 100 to 850 mg twice a day, suggesting that; the higher dose of metformin, the higher risk of lactic acidosis [14]. The therapeutic plasma concentrations of metformin measured in previous studies of type 2 diabetes ranged from 0.129 to 90 mg/L [9]. Therefore, 1 mM (129.2 mg/L) was set as a maximal concentration of metformin for our experiment, enabling the maximum achievable plasma concentration in a clinical setting without the risk of lactic acidosis.

MPA dose of 10uM was set based on a study of Zhang et al. [12] which also had evaluated the anticancer effect of MPA and metformin combination in endometrial cancer cells. MPA 10uM was high enough to inhibit proliferation of Ishikawa cells. However, it was too low to suppress progestin-resistant Ishikawa cells, which were considered to have similar characteristics with KLE cells in our study. Progestin resistance of progestin-resistant Ishikawa cells was overcome by the addition of metformin to MPA 10uM [12]. KLE and USPC cells were not expected to be susceptible to higher dose of MPA because of negative expression of ER and PR. Therefore, 10uM was set as an optimal dose of MPA which could show possible anticancer effects when combined with metformin in these cell lines. There was another study of progestin and metformin in endometrial cancer showed that 10uM MPA was the minimal dose that could significantly inhibit growth of RL95-2 cells (ER+/PR+) [15].

Can the authors better explain the AMPKalfa pathway and its function in cell proliferation?

  • Response: thank you for your comment. We added a brief explanation on AMPK pathway together with its function in cell proliferation at discussion section.

Can the authors better explain how the evaluate cell proliferation?

  • Response: thank you for your comment. As you recommended, we updated relevant part of methods section with more details. Thank you again.

Could the decrease in progesterone receptors induce resistance to MPA?

  • Response: yes, it was previously shown in the study of Xie et al. (reference 20 in the revised version). We also briefly mentioned that in our manuscript. Thank you for your comment.

Round 2

Reviewer 2 Report

Please note my previous comment about applying multiple testing correction to statistical analyses. The authors have performed multiple t-tests on the same data in Figures 1C-E, 3B-D, 4, 5 and 6. The more statistical tests are performed, the more likely a significant result will be found. It is therefore important when performing multiple tests to apply correction (i.e. adjusting of p-values) - common methods include Bonferroni (controls the familywise error rate) or Benjamini-Hochberg (controls false discovery rate) adjustment. Please state what method of multiple testing correction was applied.

Whilst revising the manuscript, the authors have introduced some new errors:

Line 25 states the “inhibitory effect of metformin was reversed when 10 µM metformin was combined” – please correct.

Lines 91-94 contradict data presented in Figure 1 - Ishikawa cells are sensitive to MPA.

Line 147: There is a random “g” at the beginning of the sentence.

Lines 222-224: Are the authors referring to KLE cells in this sentence?

Lines 362-364: I don’t understand this sentence, the second half is especially confusing – could the authors please clarify its meaning?

Figure 2: Please note that p-AMPKa should be normalised to AMPKa, whilst AMPKa, PR and ErbB2 should be normalised to GAPDH.

Figure 6: The authors need to show the effects of MMP and TGFb inhibitors alone on cell invasion. It is noted that this is mentioned in lines 364-365, but without these data the authors cannot draw any conclusion about mechanisms.

Author Response

Please note my previous comment about applying multiple testing correction to statistical analyses. The authors have performed multiple t-tests on the same data in Figures 1C-E, 3B-D, 4, 5 and 6. The more statistical tests are performed, the more likely a significant result will be found. It is therefore important when performing multiple tests to apply correction (i.e. adjusting of p-values) - common methods include Bonferroni (controls the familywise error rate) or Benjamini-Hochberg (controls false discovery rate) adjustment. Please state what method of multiple testing correction was applied.

  • Response: thank you for your valuable comments on this point. I completely understand what you mean and performed Bonferroni correction, finally confirming all Bonferroni corrected p-values were <0.05. We described this adjusting of p-values using Bonferroni correction method at method section and figure legends in Figures 1C-E, 3B-D, 4, 5 and 6. Thank you again.

Whilst revising the manuscript, the authors have introduced some new errors:

Line 25 states the “inhibitory effect of metformin was reversed when 10 µM metformin was combined” – please correct.

  • Response: thank you for your comment. We corrected this sentence replacing 10 µM metformin with 10 µM MPA.

Lines 91-94 contradict data presented in Figure 1 - Ishikawa cells are sensitive to MPA.

  • Response: a significant inhibition of Ishikawa cell proliferation after 10 µM MPA treatment in reference [12] was results at 48 hours, which was consistent with the results of our study (figure 1C). There was no data at 24 hours in reference [12]. Thus, data of figure 1C do not appear contradictory to those of reference [12]. We revised the sentence of line 91 accordingly. Thank you for your comment.

Line 147: There is a random “g” at the beginning of the sentence.

  • Response: I am sorry for this mistake. I erased “g” from the manuscript.

Lines 222-224: Are the authors referring to KLE cells in this sentence?

  • Response: no, this sentence is for Ishikawa cells. In figure 3B, the inhibitory effect of metformin and MPA combination disappeared when metformin dose was up to 1000 µM. Thus, we made no further correction on this sentence.

Lines 362-364: I don’t understand this sentence, the second half is especially confusing – could the authors please clarify its meaning?

  • Response: we revised those sentences and clarify the meaning. Thank you for your comment.

Figure 2: Please note that p-AMPKa should be normalised to AMPKa, whilst AMPKa, PR and ErbB2 should be normalised to GAPDH.

  • Response: thank you for your comment. We revised graphs of band quantification using ImageJ according to your comment. We added the following comment at figure 2 legend, “PR, ErbB2, and AMPKα were normalized to GAPDH, whilst p-AMPKα was normalized to AMPKα. “, and also revised relevant description in results section.

Figure 6: The authors need to show the effects of MMP and TGFb inhibitors alone on cell invasion. It is noted that this is mentioned in lines 364-365, but without these data the authors cannot draw any conclusion about mechanisms.

  • Response: thank you for your comment on this point. In addition to addressing this point as a limitation in discussion section, we changed the relevant expressions across the manuscript using “associated with” instead of “mediated through”.